# An evaluation of the Live Well Erie workforce development program, jumping off the benefits cliff: A protocol paper

**Christopher St. Vil**[1,2]*, **Wooksoo Kim**[1,2], **Candra Skrzypek**[1], **Meschelle Linjean**[1], **Marie A. Cannon**[1,3], **Jacqueline Hall**[1,3]

1 University at Buffalo School of Social Work, Buffalo, New York, United States of America, 2 Buffalo Center for Social Research, Buffalo, New York, United States of America, 3 Erie County Department of Social Services, Buffalo, New York, United States of America

* cstvil@buffalo.edu

**Data Availability Statement:** Deidentified research data will be made publicly available when the study is completed and published.

## Abstract

While research has suggested that addressing multiple needs simultaneously may increase the chances for positive labor force outcomes, few studies have explored a multi-pronged approach to assist individuals with jumping off the benefits cliff. The purpose of the current study is to explore the potential of a Career Pathway program designed to supplement benefits for those who "jump off the benefits cliff." This study will employ qualitative impact assessment protocol (QuIP) and observational methods to assess the implementation of the career pathway program through (1) observation, (2) focus groups with service providers and program stakeholders, and (3) semi-structured interviews with program participants. Qualitative thematic analysis will be employed to analyze responses disaggregated by the various service providers and program participants. Codes and themes generated from the analyses will be compared both within the groups of service providers and participants and between the groups.

## Introduction

Career pathway (CP) programs have been viewed as a promising approach to help low-income adults on public assistance and with limited education obtain jobs and advance up the career ladder. The central idea of CP programs is that career advancement and skill acquisition should be organized as a series of structured, successive steps accompanied by strong supports and connections to employment. The structure provides opportunities for individuals to transition to successively higher levels of credential-bearing training and employment. While some CP programs offer comprehensive services including employment and financial income support [1–3] programs vary in the type of support they provide.

Many CP programs offer career counseling or coaching to participants including counseling for personal/life challenge matters [1,4]. Some programs offer childcare and/or transportation assistance (Prins & Clymer, 2018), although outside referrals are often made for these services. Help with food, housing, utilities, healthcare, and legal aid are also sometimes

**Funding:** This work was partially supported by the program evaluation grant for the Live Well Erie Workforce Development Pilot Program from the Erie County Department of Social Services in New York award # CED 5690. The funders played no role in the design of the study protocol, however they made the final decision for publication and assisted in the preparation of the manuscript.

**Competing interests:** The authors have declared that no competing interests exist.

provided [1,4–6]. Financial support is offered in most programs which often take the form of tuition assistance, and support for fees and materials [4,6,7]. Programs also provide cash support to participants, either to cover program participation barriers [1,6] or as completion incentives [7]. However, it is not clear what combination of service offerings achieve the best outcomes for participants. The results of one program evaluation found that program participants who received both employment and financial coaching had higher net income increases than those receiving only financial or income support coaching [3].

Related to CP programs is the cliff effect, also known as the benefits cliff–which produces a major stumbling block in the path to economic mobility. The cliff effect occurs for low-income families when small increases in income result in a reduction or a loss of essential public benefits. As a result, families do not have the resources required to meet even basic needs. Earning a little more money may not automatically increase their standard of living if it boosts their income to the point where they lose access to some or all those benefits. That's because the value of those lost benefits may outweigh their income gains. As a result, low-paid workers can become reluctant to earn more money due to a fear that they will get worse off financially instead of better.

Several solutions have been recommended to address cliff effects. These solutions typically involve offering increased resources to public assistance recipients or making policy changes to public assistance programs [8]. Resource-based solutions include training caseworkers to discuss benefits cliffs with clients, offering financial literacy training to program recipients, and using calculator tools to better understand the cliff effect. Cliff-mitigating policies might include increased flexibility in reporting (e.g., providing a grace-period for re-certification deadlines), instituting benefits phase-outs with graduated eligibility criteria for programs and/ or gradual declines in assistance, eliminating asset tests for programs, and raising eligibility limits [8]. Regardless of the approach, any program seeking to enhance the skills of low-skilled workers will have to consider one or both solutions.

Recent analyses of the benefits cliff landscape in New York State found that, while New York State has enacted numerous policies to mitigate the cliff effect, other gaps in the benefits system create challenges for individuals that prevent them from reaching self-sufficiency. These gaps, collectively known as "financial gaps," include the eligibility gap, coverage gap, and hardship gap [9]. The eligibility gap occurs when one's earnings are above a benefit's eligibility threshold, yet below what is needed to meet their basic needs. For example, in New York State, an estimated 76% of households below the self-sufficiency standard do not receive food assistance. Coverage gaps occur when individuals qualify for benefits, yet do not receive them due to bureaucratic barriers, waitlists, or lack of funding. This is a particular problem with the child subsidy program as it is massively underfunded. Finally, hardship gaps occur when individuals or families receive public benefits, yet still do not have enough resources to meet their basic needs. In Erie County, where this study takes place, 27% of households earn an income above the federal poverty line yet are unable to meet their basic needs [10]. Hardship gaps often lead to benefit cliffs as individuals strive to increase their income, yet lose significant benefits (FPWA, 2021). Collectively, the cliff effect and financial gaps underscore the way that the social safety net does not allow individuals to reach self-sufficiency or economic mobility.

A recent metanalysis of 46 impact evaluations of programs that utilize elements of the CP approach found that compared to control/comparison groups, CP program participants gain a 28-percentage point advantage in earning an educational credential, a 19-percentage point advantage in obtaining employment specific to the industry they train for, and small earnings gains in the short term [11]. Prior research has demonstrated that compared to control group members, CP program participants have more consistent work hours [1,4], greater increases in wages [1,4,12], and are more likely to receive employment benefits [12]. CP participants are

also more likely to obtain a certificate or licensure and work in the profession for which they receive training [4,7]. In some studies, gains were sustained from six months [1] to six years [4] after program completion. Additionally, the participants who spent the most time utilizing all the program components had the highest job placement and six-month retention rates. A more recent process evaluation of nine programs employing "career pathway" strategies for low-income and low-skilled adults, 18 months after random assignment into treatment and control groups, is showing promising results [13]. While most of the programs being evaluated appear to be on-track to achieving their long-term goals, program participants are currently experiencing increases in average rate of employment and average earnings over successive follow-up quarters. These results suggest that multicomponent programs that address multiple participants' needs simultaneously may be particularly effective. However, while studies have explored the impact of CP programs, few have done so with an explicit emphasis on individual day to day experiences after jumping off the benefit cliff.

Prior evaluations of CP programs have primarily relied on quantitative methods such as randomized controlled trials (RCTs) with participant follow-up ranging anywhere between 18 months to 6 years later [1,4,12]. Qualitative methods were used less frequently, and primarily for studying program implementation. For example, some studies [1,5,6,12] included interviews with program staff and other stakeholders (e.g., instructors, employers), conducted during site visits, to evaluate aspects of program implementation. However, few, with the exception of [14], conducted qualitative interviews with program participants to examine their experiences in-depth. Their study interviewed a subsample of randomly assigned treatment and control group participants (n = 123), about their backgrounds, program enrollment motivations, goals, challenges experienced or expected in achieving the goals, and program experiences.

Despite the benefits of CP programs, these programs do not always consider the realities of participants who enter the program. Program retention barriers may be higher for participants who begin CP programs while receiving public benefits [5], and limited time and resources as well as life challenges increase the likelihood for program attrition [14]. Furthermore, many CP program participants express concern that increased income from the jobs they obtain through their programs will not offset the loss of public benefits such as TANF and subsidized childcare [5].

While structural changes are needed to address the root causes of financial gaps and cliff effects, there is a critical need to provide immediate support to participants faced with these challenges when designing CP programs. In many programs, when direct financial support was provided to program participants, it was in the form of a stipend or incentive [1,6,7] and not intended to offset reductions in public benefits due to increased income. Also, while some programs helped eligible participants access public benefits, they did not explicitly provide supplemental income support to fill gaps as participants inched closer to the benefits cliff. This is a critical gap to address. The purpose of the current pilot study is to explore the potential of a CP program designed to supplement benefits for those who seek to "jump off the benefits cliff," in combination with career coaching and life coaching. This approach differs from previous strategies that sought to extend eligibility for benefits and seeks to support families through supplementing financial assistance that is lost or reduced because of wage increases.

## Materials and methods

### Aims and objectives

The Erie County Department of Social Services (ECDSS) was awarded $10 million in 2021 to pilot the Live Well Erie Work Force Labor Development Program (LWE-WLDP). The

LWE-WFLDP was developed to address the needs of low-income workers through the upskilling of their skills which would lead to career advancement/higher paying positions. Specifically, the LWE-WLDP seeks to address the challenges low-income workers face as they approach the "benefits cliff" or the period when workers' wages increase to such an extent that it causes them to lose eligibility for public assistance programs. The two-year pilot program utilizes a comprehensive approach to support individuals who are currently working, receiving social service benefits, and facing the benefits cliff transition into higher-paying positions and life-long careers. Individuals who are enrolled in the program have access to 1) a career coach, 2) a life coach and 3) supplemental financial support. Supplemental financial support will be used to meet their needs for food, housing, childcare, healthcare and other hardships that may arise because of the reduction of their benefits.

The overarching goal of this evaluation is to explore the impact of the LWE-WLDP on ECDSS recipients who choose to pursue a higher-paying position and leap off the benefits cliff. This proposed research will provide a unique opportunity to explore the impact of a CP approach designed to mitigate the challenges associated with leaping off the benefits cliff. We specifically seek to (1) explore how the program was received by program recipients and program stakeholders, and (2) ascertain the impact of the loss of income due to wage increases and the role of the supplemental services of the LWE-WFLDP in mitigating the impact of the lost income. An improved understanding of how to assist low-income individuals in achieving sustainable financial gains within a context of stagnant wages, regressive tax policies, steady increases in health and education costs, ever-increasing costs of living, and the polarizing experiences of upward mobility disaggregated by class is critical. This pilot study addresses this need and is among the first of its kind undertaken in Erie County. The findings will be translated into policy recommendations to contribute to subsequent workforce labor development efforts targeted at low-income populations. Ethical approval for this study was obtained from the Institutional Review Board (IRB) at the University at Buffalo, under approval number STUDY00008160. The study complies with all institutional and national guidelines for ethical research and participants agree to participation in the study via verbal consent.

## Study approach

Exploring and mapping how a social service recipient matriculates through the program and identifying obstacles and challenges to implementation will rely on qualitative approaches. The study seeks to assess the implementation of the program through observations of staff training and monthly case reviews as well as focus groups with program staff (DSS workers, job and life coaches). The impact of the program on meeting the needs of individuals jumping off the benefits cliff will be assessed qualitatively through semi-structured interviews with program participants.

The use of a qualitative method of inquiry to assess the impact of the intervention on reduced wages is conceptually consistent with the Qualitative Impact Assessment Protocol (QuIP) approach [15]. QuIP is an impact evaluation approach that gathers evidence through narrative causal impacts collected directly from program participants, without a control group. Respondents are asked to talk about the main changes that transpired in their lives during a specified period. And prompted to share what they perceive to be the main driver of either challenges or opportunities and to whom or what they attribute those forces. For the purposes of this study, this approach recognizes the need for impact evaluations to be viewed as a social process in specific contexts [15]. As a result, this approach differs from that of [14] in that while Seefeldt and colleagues looked to interview participants within 6 months of their randomization date, this pilot seeks to interview program participants every other month

beginning with their entry into the program to attempt to generate a more longitudinal description of the proximal challenges faced by program participants as a result of diminishing benefits. This approach seeks to do the same with the program staff by observing monthly case reviews and documenting the challenges to implementation and how those challenges are addressed on a monthly basis by program stakeholders.

## Recruitment & screening

This pilot project is concerned with helping social service recipients move to self-sufficiency and transition off social services. Considering this goal, the participants who are more likely to terminate benefits or jump off the benefits cliff will be those who are already gainfully employed, motivated to stop receiving public assistance, and are already on a path to career advancement. Employers are better positioned than others to identify those among their work-force who they feel may be appropriate to enter career pathways that may lead to higher earn-ings. Employers identify workers who they deem appropriate to enter the program and explain the benefits of completing the program which include training/support and promotion after program completion. Those employees who are identified by their employers and express interest in participating in the project are then screened by ECDSS to verify service recipient status. Upon verification of status, ECDSS provides a much more in-depth description of the project. Participants are made aware that the purpose of the study is to pilot an intervention that will enable them to deal with the challenges associated with the reduction of their benefits because of their anticipated increase in earnings. After describing the intervention and the supports that will be available to the participant ECDSS solicits participant consent to enter the pilot study. This approach to sampling reflects a convenience sample approach combined with snowball sampling which is generally employed when a population is difficult to access and when non-probability sampling is not feasible. The recruitment by the employers and the screening by ECDSS will enhance the representativeness of the sample. Additionally, advertis-ing materials have been developed and widely distributed to the participating employers as well as local ECDSS offices to serve as additional modes of promotion for the pilot project.

## Eligibility criteria

Inclusion criteria for participation in the pilot study include: (1) being a current recipient of any DSS services (public assistance), (2) being currently employed with a qualified employer, (3) being referred to the program by an employer, and (4) being at least 18 years of age and able to provide consent. Qualified employers include employer organizations that ECDSS has partnered with as potential employment sites for the pilot study. These employer partners were selected because they were identified as industries (manufacturing, health care, hospital-ity, business/customer service, and technology) that promoted pathways to career advance-ment and were willing to partner with DSS to achieve the goals of the program.

## Sample size

A pragmatic model to assess sample size in qualitative research has been proposed that consid-ers five factors: 1) study aim, 2) sample specificity, 3) established theory, 4) of dialogue, and 5) analysis strategy [16]. The model dictates that considerations about study aim, sample specific-ity, theoretical background, quality of dialogue, and strategy for analysis should determine whether sufficient information power will be obtained with less or more participants included in the sample. While the phenomenon of those seeking to enter career pathway programs from a starting point of unemployment is broad, the focus on those who are already employed and seeking to "jump off" the benefits cliff requires a much narrower study aim and sample

specificity. Given the reluctance of individuals receiving benefits to "jump off" the cliff, the aim of the study concerns a rare experience that results in a limited number of eligible participants and a higher level of information power. An established framework for understanding the experiences of those attempting to jump off the benefits cliff is rooted in previous literature. Applying that literature as the starting point for understanding the experiences that program participants share during interviews will reduce the need for larger sample sizes. Given that the proposed model necessitates a small sample, we propose to interview 30–50 program participants. All program staff, however, including DSS workers, career coaches, and life coaches will be included in the stakeholder interviews.

## Study procedure

### Observation (implementation)

Researchers will observe program processes in light of service provider training as well as monthly case reviews.

**Service provider training.** Those involved in the provision of services to study participants (i.e., DSS workers, Career and Life coaches) will participate in an intensive training program prior to the start of data collection. The training addressed their role as service providers, the tools they have at their disposal (i.e., benefits calculators, the release of supplemental services), and how to access the newly developed supplemental funds on behalf of those participants experiencing a reduction in their wages. The researchers will attend the training as participant observers and take notes during the training. Additionally, questionnaires will be distributed to the stakeholders to acquire service provider feedback about the quality and usefulness of the training.

**Monthly case reviews.** A case review. For the purposes of this study, is an inherently person-centered process, used to assist in managing, coordinating, and reviewing responses to what is happening in the life of a participant that may prevent them from acquiring a higher wage or jumping off the benefits cliff. It is an opportunity to support a program participant by bringing together multiple stakeholders (i.e., career and life coaches and DSS workers) who are jointly well-placed to understand and address the program participants' holistic needs and situation. Case reviews will take place monthly and consist of a review of all participating program participants. The researchers will attend the monthly reviews and audio-record them for analysis. The observation of the monthly case reviews will highlight the day-to-day issues program participants are encountering as obstacles to their goals.

### Focus groups (implementation)

In addition to monthly case reviews with program service providers to identify program challenges, separate focus groups will be conducted with (1) DSS workers, (2) career coaches, and (3) life coaches. The groups are distinguished to allow discussion around programmatic issues to revolve and pertain to their respective group. For example, the life coaches are welcomed to one focus group so that the issues relevant to them are heard and addressed and do not intersect with issues stemming from those impacting DSS workers. These groups with program service providers will take every three months.

### Semi-structured interviews with program participants (outcome)

Program participants will be invited to engage in semi-structured qualitative interviews every two months. The goal of the interviews is to ascertain from the perspective of the program participants, the status of their progress, challenges, and their general outlook toward their

trajectory. Interviewing the participants every two months will allow the researchers to document the proximate challenges individuals approaching the benefits cliff experience throughout the process and that are routinely missed when researchers focus on interviews toward the end of a participant's experience with a program. The focus here is on describing the process of adjusting to decreased wages over time as opposed to focusing on the outcome.

## Intervention

**Wraparound services.** Wraparound services consist of the coupling of the study participants with both a career coach and a life coach. The career coach advises the participant on job related issues such as employment etiquette and resume development. The career coach serves as a support to help develop the non-tangible skills needed in the workplace to further progress a career pathway. The life coach in contrast seeks to support the participant around issues that may serve as barriers to completely jumping off the benefit cliff. For example, to help allay the fears associated with the termination of benefits, life coaches may help participants develop a budget or help them identify alternative strategies to navigate the period when benefits will end. Much of their work is psychological and focused on increasing adaptability and reducing fear. These two pillars of support, the life and career coach used simultaneously, to support the participant over the benefits cliff is the unique feature of this intervention. These two actors, along with the ECDSS worker will work together to form the wraparound services core on behalf of the participant.

**Upskilling opportunities.** The upskilling of participants or the acquisition of new skills will take place in two ways. First, participating employers will provide on-the-job training that will lead to program participants developing a career pathway that will lead to a higher-paying position within a year of entry into the pilot project. Employers select employees they feel would be a good match and/or benefit the most from the career pathway opportunity. One example of this upskilling is the provision of English classes on site at the job to serve as a resource for ESL learners to improve their communication skills. Second, the support provided by the career coaches will supplement additional skill acquisition. This will be done through individual case management and referrals to training opportunities that align with the career pathway of a participant.

**Supplemental funds.** The third and final component of the intervention is the provision of supplemental funds. The purpose of the supplemental funds is to (1) temporarily replace the funds lost because of a wage increase and (2) aid in the event of an emergency that would lead to the participant being terminated or being forced to resign had the funds not been available. These supplemental financial resources were developed to ease the transition of those jumping off the cliff. Career and life coaches have the authority to request these funds on behalf of a program participant. However, the funds must be approved by ECDSS prior to disbursement.

## Criteria for discontinuing or modifying intervention

Due to this being a pilot, we reasonably expect to modify the intervention as the program progresses. Weekly meetings with program stakeholders and bi-weekly meetings with program administrators serve as artifacts of data for program stakeholders to determine how best to modify the intervention to best serve the needs of the program participants.

## Data analysis

Descriptive, univariate statistics will be generated that describe the characteristics of the participant and staff sample. Demographic variables collected for service providers include gender role and tenure of current position. Demographic variables collected for participants will

include race, gender, length of time receiving benefits, type of benefits received, length and type of employment, number of children, and vehicle ownership. Efforts will be made when appropriate to disaggregate the samples by varying characteristics.

All qualitative data including focus groups, monthly case reviews, and semi-structured interviews will all be transcribed and uploaded into Atlas TI software. The analyses will be conducted separately according to service provider to allow for the isolation and analyses of experiences by service provider role (i.e., life coach, career coach, DSS worker) and participant. In this manner, the experiences of all four parties can be compared, contrasted, and juxtaposed. The service provider (life coaches, career coaches, DSS workers) transcripts focus on implementation challenges and program fidelity and the participant transcripts focus on daily challenges of wage reduction and aspects of the intervention they found helpful. The qualitative analysis will take place in three stages across the four groups of respondents. Relying on an inductive logic approach, the analyses will begin with an initial reading of all transcripts by the research team to generate rudimentary, initial codes. This first round of coding represents the open coding stage. A second round of analyses will consist of the pairing of the initial codes with their associated text/themes. The second round compiles the themes across respondents according to each generated code. Themes are then analyzed exclusively within each service provider group based on role and the participants, respectively. Consequently, themes and patterns of responses are generated for each group of respondents. In the third round, the themes generated from each group will be compared across the mutually exclusive groups. Generated themes from both the within-group and between-group comparisons will be discussed in relation to the objectives of the study and how they converge lending the process to a form of triangulation. The analysis of the data will continue until saturation is reached.

## Data management and privacy

Data will be collected through an electronic data collection portal developed specifically for this pilot study. The portal is maintained by ECDSS. As a result, all demographic data as well as type of benefits and the tenure of benefit receipt is pre-entered into the data collection portal for each individual participating in the study. The database was also developed in a manner to support qualitative data so that case notes written by coaches can be written into a tablet computer and directly uploaded to the data collection portal. Access to the data is limited to the evaluators and DSS workers. DSS stakeholders and the evaluation partners have passwords to the portal. Data for analysis is provided to the evaluators through "data dumps" which are de-identified and converted to Excel files.

## Confidentiality

All program data shared with the program evaluators will be de-identified and have no identifying information attached to the data files. ECDSS will delete identifying demographic information from the dataset and replace the names of program participants with pseudonyms.

## Compensation

Participants will receive a $30.00 gift card for their participation.

## Potential risks

One risk to this pilot study includes the possible breach of data. Data is never 100% protected but the risks of data breach are mitigated by adhering to the policies and procedures of the data collection and safe storage policies.

### Potential benefits

Participation in this pilot study will contribute to an overall understanding of the challenges to jumping off the benefits cliff when experiencing a rise in wages. More importantly, this pilot study has the potential to inform work labor force policies in Western NY. If positive, the findings from this pilot study could persuade policy makers to support a wider expansion of the project.

### Dissemination

The research team plans to communicate the results of the study to participants and relevant stakeholders in the county of Erie and various stakeholders associated with work force labor operations on a county level. Additional dissemination will proceed through the submission of peer reviewed manuscripts, conference presentations as well as local presentations to stakeholders in work force development at the state and city level.

## Discussion

This protocol describes the examination of an intervention that seeks to mitigate workforce labor development challenges to benefit recipients who are seeking to jump off the benefits cliff. The pilot study is unique in a few ways. First, this intervention is not focused on helping unemployed persons find employment. Rather, this intervention is concerned with assisting benefit recipients that are already employed improve their earnings and ultimately terminate benefits. Second, the methodological approach employed here departs from the traditional quantitative analysis and seeks to emphasize patient narratives that highlight the day to day struggles that prevent persons from terminating benefits. Specifically, while previous qualitative work has interviewed program participants at the end of an intervention, our approach seeks to interview them multiple times throughout the tenure of the study to ascertain and track challenges over time. While structural changes are needed to address the root causes of financial gaps and cliff effects, there is a critical need to provide immediate support to participants faced with these challenges when designing CP programs. Lastly, this pilot study is particularly interested in examining how the three interventions (wraparound services, upskilling, and supplemental funds) either impede or help program participants end their relationship with ECDSS. This pilot study is purely qualitative and in line with the QuIP approach.

Public benefit recipients who experience increases in wages are financially worse off in the short-term and medium-term to an extent that little incentive exists for them to continue moving up the pay ladder. However, the long-term gains to both the community and the individual over time result in potential net saving to taxpayers. The results of our study will inform strategies to address the challenges of benefit recipients who are already employed, thereby promoting financial independence.

## Author Contributions

**Conceptualization:** Christopher St. Vil, Wooksoo Kim, Marie A. Cannon, Jacqueline Hall.

**Writing – original draft:** Christopher St. Vil, Candra Skrzypek, Meschelle Linjean.

**Writing – review & editing:** Christopher St. Vil, Candra Skrzypek, Meschelle Linjean, Marie A. Cannon, Jacqueline Hall.

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
