## [Decision Letter · Decision Letter 0]

14 Aug 2024

PONE-D-24-24302An Evaluation of the Live Well Erie Work Force Labor Development Program, Jumping Off the Benefits Cliff:  A Protocol PaperPLOS ONE

Dear Dr. St. Vil,

Thank you for submitting your manuscript to PLOS ONE. After careful consideration, we feel that it has merit but does not fully meet PLOS ONE’s publication criteria as it currently stands. Therefore, we invite you to submit a revised version of the manuscript that addresses the points raised during the review process.

We look forward to receiving your revised manuscript.

Kind regards,

Vanessa Carels

Staff Editor

PLOS ONE

2. PLOS requires an ORCID iD for the corresponding author in Editorial Manager on papers submitted after December 6th, 2016. Please ensure that you have an ORCID iD and that it is validated in Editorial Manager. To do this, go to ‘Update my Information’ (in the upper left-hand corner of the main menu), and click on the Fetch/Validate link next to the ORCID field. This will take you to the ORCID site and allow you to create a new iD or authenticate a pre-existing iD in Editorial Manager. Please see the following video for instructions on linking an ORCID iD to your Editorial Manager account: https://www.youtube.com/watch?v=_xcclfuvtxQ".

 [Erie County Department of Social Services].  

6. Please amend your list of authors on the manuscript to ensure that each author is linked to an affiliation. Authors’ affiliations should reflect the institution where the work was done (if authors moved subsequently, you can also list the new affiliation stating “current affiliation:….” as necessary).

7. Please amend either the abstract on the online submission form (via Edit Submission) or the abstract in the manuscript so that they are identical.

8. Please include your full ethics statement in the ‘Methods’ section of your manuscript file. In your statement, please include the full name of the IRB or ethics committee who approved or waived your study, as well as whether or not you obtained informed written or verbal consent. If consent was waived for your study, please include this information in your statement as well.

Reviewers' comments:

Reviewer's Responses to Questions

**Comments to the Author**

1. Does the manuscript provide a valid rationale for the proposed study, with clearly identified and justified research questions?

Reviewer #1: Yes

2. Is the protocol technically sound and planned in a manner that will lead to a meaningful outcome and allow testing the stated hypotheses?

Reviewer #1: Yes

3. Is the methodology feasible and described in sufficient detail to allow the work to be replicable?

Reviewer #1: Yes

4. Have the authors described where all data underlying the findings will be made available when the study is complete?

Reviewer #1: Yes

5. Is the manuscript presented in an intelligible fashion and written in standard English?

Reviewer #1: Yes

6. Review Comments to the Author

You may also provide optional suggestions and comments to authors that they might find helpful in planning their study.

Reviewer #1: Study protocol clearly presented. Knowledge gaps identified. The authors might consider strengthening the literature review. There is an extensive literature on career pathways. More depth is needed.

7. PLOS authors have the option to publish the peer review history of their article (what does this mean?). If published, this will include your full peer review and any attached files.

Reviewer #1: No

---

## [Author Response · Author response to Decision Letter 0]

15 Sep 2024

Response to Reviewers

We apologize for not adhering to the PLOS ONE style requirements during the initial submission. We have modified the manuscript to adhere to your publication criteria. We followed the style templates through the links you provided and modified the headings within the document and cited our references within brackets.

2. PLOS requires an ORCID iD for the corresponding author in Editorial Manager on papers submitted after December 6th, 2016. Please ensure that you have an ORCID iD and that it is validated in Editorial Manager.

The Orcid ID for the corresponding author has been included. https://orcid.org/0000-0003-3501-2530

The discrepancy between the information in the funding information and financial disclosure sections has been rectified. They now match and we have ensured that the grant number is indeed correct.

 [Erie County Department of Social Services]. 

We have amended the statement in the cover letter as requested. The statement now reads, “The funders played no role in the design of the study protocol, however they made the final decision for publication and assisted in the preparation of the manuscript.”

As this is a protocol paper, we don’t have complete data to share yet. However, we will be prepared to share de-identified quantitative data at the conclusion of the study.

6. Please amend your list of authors on the manuscript to ensure that each author is linked to an affiliation. Authors’ affiliations should reflect the institution where the work was done (if authors moved subsequently, you can also list the new affiliation stating “current affiliation:….” as necessary).

The list of authors has been amended to ensure each author is linked to an affiliation.

7. Please amend either the abstract on the online submission form (via Edit Submission) or the abstract in the manuscript so that they are identical.

The abstract on the online submission form and the manuscript now match.

8. Please include your full ethics statement in the ‘Methods’ section of your manuscript file. In your statement, please include the full name of the IRB or ethics committee who approved or waived your study, as well as whether or not you obtained informed written or verbal consent. If consent was waived for your study, please include this information in your statement as well.

The ethics statement has been included and inserted in the methods section of the manuscript file.

The following reference was included to address Reviewer #1 feedback.

Gardiner, K. and R. Juras. (2019). PACE Cross-Program Implementation and Impact Study Findings, OPRE Report #2019-32, Washington, DC: Office of Planning, Research, and Evaluation, Administration for Children and Families, U.S. Department of Health and Human Services. 

10. Reviewer #1: Study protocol clearly presented. Knowledge gaps identified. The authors might consider strengthening the literature review. There is an extensive literature on career pathways. More depth is needed.

We express gratitude to the reviewer for their feedback. We incorporated the following statement on pg. 4, paragraph 2. “A more recent process evaluation of nine programs employing “career pathway” strategies for low-income and low-skilled adults, 18 months after random assignment into treatment and control groups, is showing promising results (Gardiner & Juras, 2019). While most of the programs being evaluated appear to be on-track to achieving their long-term goals, program participants are currently experiencing increases in average rate of employment and average earnings over successive follow-up quarters.”

---

## [Editor Report · Decision Letter 1]

18 Sep 2024

An Evaluation of the Live Well Erie Work Force Labor Development Program, Jumping Off the Benefits Cliff:  A Protocol Paper

PONE-D-24-24302R1

Dear Dr. St. Vil,

We’re pleased to inform you that your manuscript has been judged scientifically suitable for publication and will be formally accepted for publication once it meets all outstanding technical requirements.

Kind regards,

Vanessa Carels

Staff Editor

PLOS ONE
---

## [Editor Report · Acceptance letter]

5 Nov 2024

PONE-D-24-24302R1 

PLOS ONE

Dear Dr. St. Vil, 

I'm pleased to inform you that your manuscript has been deemed suitable for publication in PLOS ONE. Congratulations! Your manuscript is now being handed over to our production team.

Kind regards, 

on behalf of

Dr. Vanessa Carels 

Staff Editor

PLOS ONE